# Scaling Factorial Hidden Markov Models: Stochastic Variational Inference without Messages

**Yin Cheng Ng**
Dept. of Statistical Science
University College London
y.ng.12@ucl.ac.uk

**Pawel Chilinski**
Dept. of Computing Science
University College London
ucabchi@ucl.ac.uk

**Ricardo Silva**
Dept. of Statistical Science
University College London
r.silva@ucl.ac.uk

## Abstract

Factorial Hidden Markov Models (FHMMs) are powerful models for sequential data but they do not scale well with long sequences. We propose a scalable inference and learning algorithm for FHMMs that draws on ideas from the stochastic variational inference, neural network and copula literatures. Unlike existing approaches, the proposed algorithm requires no message passing procedure among latent variables and can be distributed to a network of computers to speed up learning. Our experiments corroborate that the proposed algorithm does not introduce further approximation bias compared to the proven structured mean-field algorithm, and achieves better performance with long sequences and large FHMMs.

## 1 Introduction

Breakthroughs in modern technology have allowed more sequential data to be collected in higher resolutions. The resulted sequential data sets are often extremely long and high-dimensional, exhibiting rich structures and long-range dependency that can only be captured by fitting large models to the sequences, such as Hidden Markov Models (HMMs) with a large state space. The standard methods of learning and performing inference in the HMM class of models are the Expectation-Maximization (EM) and the Forward-Backward algorithms. The Forward-Backward and EM algorithms are prohibitively expensive for long sequences and large models because of their linear and quadratic computational complexity with respect to sequence length and state space size respectively.

To rein in the computational cost of inference in HMMs, several variational inference algorithms that trade-off inference accuracy in exchange for lower computational cost have been proposed in the literatures. Variational inference is a deterministic approximate inference technique that approximates posterior distribution $p$ by minimizing the Kullback-Leibler divergence $KL(q||p)$, where $q$ lies in a family of distributions selected to approximate $p$ as closely as possible while keeping the inference algorithm computationally tractable [24]. Despite its biased approximation of the actual posteriors, the variational inference approach has been proven to work well in practice [21].

Variational inference has also been successfully scaled to tackle problems with large data sets through the use of stochastic gradient descent (SGD) algorithms [12]. However, applications of such techniques to models where the data is dependent (i.e., non-i.i.d.) require much care in the choice of the approximating family and parameter update schedules to preserve dependency structure in the data [9]. More recently, developments of stochastic variational inference algorithms to scale models for non-i.i.d. data to large data sets have been increasingly explored [5, 9].

We propose a stochastic variational inference approach to approximate the posterior of hidden Markov chains in Factorial Hidden Markov Models (FHMM) with independent chains of bivariate Gaussian copulas. Unlike existing variational inference algorithms, the proposed approach eliminates the need for explicit message passing between latent variables and allows computations to be distributed to

multiple computers. To scale the variational distribution to long sequences, we reparameterise the bivariate Gaussian copula chain parameters with feed-forward recognition neural networks that are shared by copula chain parameters across different time points. The use of recognition networks in variational inference has been well-explored in models in which data is assumed to be i.i.d. [14, 11]. To the best of our knowledge, the use of recognition networks to decouple inference in non-factorised stochastic process of unbounded length has not been well-explored. In addition, both the FHMM parameters and the parameters of the recognition networks are learnt in conjunction by maximising the stochastic lower bound of the log-marginal likelihood, computed based on randomly sampled subchains from the full sequence of interest. The combination of recognition networks and stochastic optimisations allow us to scale the Gaussian copula chain variational inference approach to very long sequences.

## 2  Background

### 2.1  Factorial Hidden Markov Model

Factorial Hidden Markov Models (FHMMs) are a class of HMMs consisting of $M$ latent variables $\mathbf{s}_t = (s_t^1, \cdots, s_t^M)$ at each time point, and observations $y_t$ where the conditional emission probability of the observations $p\left(y_t | \mathbf{s}_t, \eta\right)$ is parameterised through factorial combinations of $\mathbf{s}_t$ and emission parameters $\eta$. Each of the latent variables $s_t^m$ evolves independently in time through discrete-valued Markov chains governed by transition matrix $A_m$ [8]. For a sequence of observations $\mathbf{y} = (y_1, \cdots, y_T)$ and corresponding latent variables $\mathbf{s} = (\mathbf{s}_1, \cdots, \mathbf{s}_T)$, the joint distribution can be written as follow

$$p(\mathbf{y}, \mathbf{s}) = \prod_{m=1}^{M} p(s_1^m) p(y_1 | \mathbf{s}_1, \eta) \prod_{t=2}^{T} p(y_t | \mathbf{s}_t, \eta) \prod_{m=1}^{M} p(s_t^m | s_{t-1}^m, A_m) \tag{1}$$

Depending on the state of latent variables at a particular time point, different subsets of emission parameters $\eta$ can be selected, resulting in a dynamic mixture of distributions for the data. The factorial respresentation of state space reduces the required number of parameters to encode transition dynamics compared to regular HMMs with the same number of states. As an example, a state space with $2^M$ states can be encoded by $M$ binary transition matrices with a total of $2M$ parameters while a regular HMM requires a transition matrix with $2^M \times (2^M - 1)$ parameters to be estimated.

In this paper, we specify a FHMM with $D-$dimensional Gaussian emission distributions and $M$ binary hidden Markov chains. The emission distributions share a covariance matrix $\Sigma$ across different states while the mean is parameterised as a linear combination of the latent variables,

$$\mu_t = \mathbf{W}^T \hat{\mathbf{s}}_t, \tag{2}$$

where $\hat{\mathbf{s}}_t = [s_t^1, \cdots, s_t^M, 1]^T$ is a $M + 1$-dimensional binary vector and $\mathbf{W} \in \mathbb{R}^{(M+1) \times D}$. The FHMM model parameters $\Gamma = (\Sigma, \mathbf{W}, A_1, \cdots, A_M)$ can be estimated with the EM algorithm. Note that to facilitate optimisations, we reparameterised $\Sigma$ as $\mathbf{L}\mathbf{L}^T$ where $\mathbf{L} \in \mathbb{R}^{D \times D}$ is a lower-triangular matrix.

#### 2.1.1  Inference in FHMMs

Exact inference in FHMM is intractable due to the $O(TMK^{M+1})$ computational complexity for FHMM with $M$ $K$-state hidden Markov chains [15]. A structured mean-field (SMF) variational inference approach proposed in [8] approximates the posterior distribution with $M$ independent Markov chains and reduces the complexity to $O(TMK^2)$ in models with linear-Gaussian emission distributions. While the reduction in complexity is significant, inference and learning with SMF remain insurmountable in the presence of extremely long sequences. In addition, SMF requires the storage of $O(2TMK)$ variational parameters in-memory per training sequence. Such computational requirements remain expensive to satisfy even in the age of cloud computing.

### 2.2  Gaussian Copulas

Gaussian copulas are a family of multivariate cumulative distribution functions (CDFs) that capture linear dependency structure between random variables with potentially different marginal distributions. Given two random variables $X_1, X_2$ with their respective marginal CDFs $F_1, F_2$, their Gaussian copula joint CDF can be written as

$$\Phi_\rho(\phi^{-1}(F_1(x_1)), \phi^{-1}(F_2(x_2))) \tag{3}$$

where $\phi^{-1}$ is the quantile function of the standard Gaussian distribution, and $\Phi$ is the CDF of the standard bivariate Gaussian distribution with correlation $\rho$. In a bivariate setting, the dependency between $X_1$ and $X_2$ is captured by $\rho$. The bivariate Gaussian copula can be easily extended to multivariate settings through a correlation matrix. For an in-depth introduction of copulas, please refer to [18, 3].

### 2.3 Stochastic Variational Inference

Variational inference is a class of deterministic approximate inference algorithms that approximate intractable posterior distributions $p(\mathbf{s}|\mathbf{y})$ of latent variables $\mathbf{s}$ given data $\mathbf{y}$ with a tractable family of variational distributions $q_\beta(\mathbf{s})$ parameterised by variational parameters $\beta$. The variational parameters are fitted to approximate the posterior distributions by maximising the evidence lower bound of log-marginal likelihood (ELBO) [24]. By applying the Jensen's inequality to $\int p(\mathbf{y}, \mathbf{s})d\mathbf{s}$ the ELBO can be expressed as

$$ELBO = \mathbb{E}_q[\log p(\mathbf{y}, \mathbf{s})] - \mathbb{E}_q[\log q(\mathbf{s})]. \tag{4}$$

The ELBO can also be interpreted as the negative KL-divergence $KL(q_\beta(\mathbf{s})||p(\mathbf{s}|\mathbf{y}))$ up to a constant. Therefore, variational inference results in variational distribution that is the closest to $p$ within the approximating family as measured by $KL$.

Maximising ELBO in the presence of large data set is computationally expensive as it requires the ELBO to be computed over all data points. Stochastic variational inference (SVI) [12] successfully scales the inference technique to large data sets using subsampling based stochastic gradient descent algorithms[2].

### 2.4 Amortised Inference and Recognition Neural Networks

The many successes of neural networks in tackling certain supervised learning tasks have generated much research interest in applying neural networks to unsupervised learning and probabilistic modelling problems [20, 7, 19, 14]. A recognition neural network was initially proposed in [11] to extract underlying structures of data modelled by a generative neural network. Taking the observed data as input, the feed-forward recognition network learns to predict a vector of unobserved code that the generative neural network initially conjectured to generate the observed data.

More recently, a recognition network was applied to variational inference for latent variable models[14, 7]. Given data, the latent variable model and an assumed family of variational distributions, the recognition network learns to predict optimal variational parameters for the specific data points. As the recognition network parameters are shared by all data points, information learned by the network on a subset of data points are shared with other data points. This inference process is aptly named amortised inference. In short, recognition network can simply be thought of as a feed-forward neural network that learns to predict optimal variational parameters given the observed data, with ELBO as its utility function.

## 3 The Message Free Stochastic Variational Inference Algorithm

While structured mean-field variational inference and its associated EM algorithms are effective tools for inference and learning in FHMMs with short sequences, they become prohibitively expensive as the sequences grow longer. For example, one iteration of SMF forward-backward message passing for FHMM of 5 Markov chains and $10^6$ sequential data points takes hours of computing time on a modern 8-cores workstation, rendering SMF unusable for large scale problems. To scale FHMMs to long sequences, we resort to stochastic variational inference.

The proposed variational inference algorithm approximates posterior distributions of the $M$ hidden Markov chains in FHMM with $M$ independent chains of bivariate Gaussian-Bernoulli copulas. The computational cost of optimising the variational parameters is managed by a subsampling-based stochastic gradient ascent algorithm similar to SVI. In addition, parameters of the copula chains are reparameterised using feed-forward recognition neural networks to improve efficiency of the variational inference algorithm.

In contrast to the EM approach for learning FHMM model parameters, our approach allows for both the model parameters and variational parameters to be learnt in conjunction by maximising the ELBO

with a stochastic gradient ascent algorithm. In the following sections, we describe the variational distributions and recognition networks, and derive the stochastic ELBO for SGD.

## 3.1 Variational Chains of Bivariate Gaussian Copulas

Similar to the SMF variational inference algorithm proposed in [8], we aim to preserve posterior dependency of latent variables within the same hidden Markov chains by introducing chains of bivariate Gaussian copulas. The chain of bivariate Gaussian copulas variational distribution can be written as the product of bivariate Gaussian copulas divided by the marginals of latent variables at the intersection of the pairs,

$$q(\mathbf{s}^m) = \frac{\prod_{t=2}^{T} q(s_{t-1}^m, s_t^m)}{\prod_{t=2}^{T-1} q(s_t^m)} \tag{5}$$

where $q(s_{t-1}^m, s_t^m)$ is the joint probability density or mass function of a bivariate Gaussian copula.

The copula parameterization in Equation (5) offers several advantages. Firstly, the overlapping bivariate copula structure enforces coherence of $q(s_t^m)$ such that $\sum_{s_{t-1}^m} q(s_{t-1}^m, s_t^m) = \sum_{s_{t+1}^m} q(s_t^m, s_{t+1}^m)$. Secondly, the chain structure of the distribution restricts the growth in the number of variational parameters to only two parameters per chain for every increment in the sequence length. Finally, the Gaussian copula allows marginals and dependency structure of the random variables to be modelled separately [3]. The decoupling of the marginal and correlation parameters thus allows these parameters to be estimated by unconstrained optimizations and also lend themselves to be predicted separately using feed-forward recognition neural networks.

For the rest of the paper, we assume that the FHMM latent variables are Bernoulli random variables with the following bivariate Gaussian-Bernoulli copula probability mass function (PMF) as their variational PMFs

$$
\begin{aligned}
q(s_{t-1}^m = 0, s_t^m = 0) &= q_{00_t}^m & q(s_{t-1}^m = 1, s_t^m = 0) &= 1 - \theta_{t,m} - q_{00_t}^m \\
q(s_{t-1}^m = 0, s_t^m = 1) &= 1 - \theta_{t-1,m} - q_{00_t}^m & q(s_{t-1}^m = 1, s_t^m = 1) &= \theta_{t,m} + \theta_{t-1,m} + q_{00_t}^m - 1
\end{aligned}
\tag{6}
$$

where $q_{00_t}^m = \Phi_{\rho_{t,m}}(\phi^{-1}(1 - \theta_{t-1,m}), \phi^{-1}(1 - \theta_{t,m}))$ and $q(s_t^m = 1) = \theta_{t,m}$. The Gaussian-Bernoulli copula can be easily extended to multinomial random variables.

Assuming independence between random variables in different hidden chains, the posterior distribution of $\mathbf{s}$ can be factorised by chains and approximated by

$$q(\mathbf{s}) = \prod_{m=1}^{M} q(\mathbf{s}^m) \tag{7}$$

## 3.2 Feed-forward Recognition Neural Networks

The number of variational parameters in the chains of bivariate Gaussian copulas scales linearly with respect to the length of the sequence as well as the number of sequences in the data set. While it is possible to directly optimise these variational parameters, the approach quickly becomes infeasible as the size of data set grows. We propose to circumvent the challenging scalability problem by reparameterising the variational parameters with rolling feed-forward recognition neural networks that are shared among variational parameters within the same chain. The marginal variational parameters $\theta_{t,m}$ and copula correlation variational parameters $\rho_{t,m}$ are parameterised with different recognition networks as they are parameters of a different nature.

Given observed sequence $\mathbf{y} = (y_1, \ldots, y_T)$, the marginal and correlation recognition networks for hidden chain $m$ compute the variational paremeters $\theta_{t,m}$ and $\rho_{t,m}$ by performing a forward pass on a window of observed data $\Delta \mathbf{y}_t = (y_{t-\frac{1}{2}\Delta t}, \ldots, y_t, \ldots, y_{t+\frac{1}{2}\Delta t})$

$$\theta_{t,m} = f_\theta^m(\Delta \mathbf{y}_t) \qquad \rho_{t,m} = f_\rho^m(\Delta \mathbf{y}_t) \tag{8}$$

where $\Delta t + 1$ is the user selected size of rolling window, $f_\theta^m$ and $f_\rho^m$ are the marginal and correlation recognition networks for hidden chain $m$ with parameters $\omega_m = (\omega_{\theta,m}, \omega_{\rho,m})$. The output layer non-linearities of $f_\theta^m$ and $f_\rho^m$ are chosen to be the sigmoid and hyperbolic tangent functions respectively to match the range of $\theta_{t,m}$ and $\rho_{t,m}$.

The recognition network hyperparameters, such as the number of hidden units, non-linearities, and the window size $\Delta t$ can be chosen based on computing budget and empirical evidence. In our

experiments with shorter sequences where ELBO can be computed within a reasonable amount of time, we did not observe a significant difference in the coverged ELBOs among different choices of non-linearity. However, we observed that the converged ELBO is sensitive to the number of hidden units and the number of hidden units needs to be adapted to the data set and computing budget. Recognition networks with larger hidden layers have larger capacity to approximate the posterior distributions as closely as possible but require more computing budget to learn. Similarly, the choice of $\Delta t$ determines the amount of information that can be captured by the variational distributions as well as the computing budget required to learn the recognition network parameters. As a rule of thumb, we recommend the number of hidden units and $\Delta t$ to be chosen as large as the computing budget allows in long sequences. We emphasize that the range of posterior dependency captured by the correlation recognition networks is not limited by $\Delta t$, as the recognition network parameters are shared across time, allowing dependency information to be encoded in the network parameters. For FHMMs with large number of hidden chains, various schemes to share the networks' hidden layers can be devised to scale the method to FHMMs with a large state space. This presents another trade-off between computational requirements and goodness of posterior approximations.

In addition to scalability, the use of recognition networks also allows our approach to perform fast inference at run-time, as computing the posterior distributions only require forward passes of the recognition networks with data windows of interest. The computational complexity of the recognition network forward pass scales linearly with respect to $\Delta t$. As with other types of neural networks, the computation is highly data-parallel and can be massively sped up with GPU. In comparison, computation for a stochastic variational inference algorithm based on a message passing approach also scales linearly with respect to $\Delta t$ but is not data-parallel [5]. Subchains from long sequences, together with their associated recognition network computations, can also be distributed across a cluster of computers to improve learning and inference speed.

However, the use of recognition networks is not without its drawbacks. Compared to message passing algorithms, the recognition networks approach cannot handle missing data gracefully by integrating out the relevant random variables. The fidelity of the approximated posterior can also be limited by the capacity of the neural networks and bad local minimas. The posterior distributions of the random variables close to the beginning and the end of the sequence also require special handling, as the rolling window cannot be moved any further to the left or right of the sequences. In such scenarios, the posteriors can be computed by adapting the structured mean-field algorithm proposed in [8] to the subchains at the boundaries (see *Supplementary Material*). The importance of the boundary scenarios in learning the FHMM model parameters diminishes as the data sequence becomes longer.

### 3.3 Learning Recognition Network and FHMM Parameters

Given sequence $\mathbf{y}$ of length $T$, the $M$-chain FHMM parameters $\Gamma$ and recognition network parameters $\Omega = (\omega_1, \ldots, \omega_M)$ need to be adapted to the data by maximising the ELBO as expressed in Equation (4) with respect to $\Gamma$ and $\Omega$. Note that the distribution $q(\mathbf{s}^m)$ is now parameterised by the recognition network parameters $\omega_m$. For notational simplicty, we do not explicitly express the parameterisation of $q(\mathbf{s}^m)$ in our notations. Plugging in the FHMM joint distribution in Equation (1) and variational distribution in Equation (7), the FHMM ELBO $\mathbb{L}(\Gamma, \Omega)$ for the variational chains of bivariate Gaussian copula is approximated as

$$
\mathbb{L}(\Gamma, \Omega) \approx \sum_{t=\frac{1}{2}\Delta t+1}^{T-\frac{1}{2}\Delta t-1} \big\langle \log p(y_t|s_t^1, \ldots, s_t^M) \big\rangle_q
$$

$$
+ \sum_{m=1}^{M} \big\langle \log p(s_t^m|s_{t-1}^m) \big\rangle_q + \big\langle \log q(s_t^m) \big\rangle_q - \big\langle \log q(s_t^m, s_{t+1}^m) \big\rangle_q \tag{9}
$$

Equation (9) is only an approximation of the ELBO as the variational distribution of $s_t^m$ close to the beginning and end of $\mathbf{y}$ cannot be computed using the recognition networks. Because of the approximation, the FHMM initial distribution $\prod_{m=1}^{M} p(s_1^m)$ cannot be learned using our approach. However, they can be approximated by the stationary distribution of the transition matrices as $T$ become large assuming that the sequence is close to stationary[5]. Comparisons to SMF in our experiment results suggest that the error caused by the approximations is negligible.

The log-transition probability expectations and variational entropy in Equation (9) can be easily computed as they are simply sums over pairs of Bernoulli random variables. The expectations of

log-emission distributions can be efficiently computed for certain distributions, such as multinomial and multivariate Gaussian distributions. Detailed derivations of the expectation terms in ELBO can be found in the *Supplementary Material*.

### 3.3.1 Stochastic Gradient Descent & Subsampling Scheme

We propose to optimise Equation (9) with SGD by computing noisy unbiased gradients of ELBO with respect to $\Gamma$ and $\Omega$ based on contributions from subchains of length $\Delta t + 1$ randomly sampled from $\mathbf{y}$ [2, 12]. Multiple subchains can be sampled in each of the learning iterations to form a mini-batch of subchains, reducing variance of the noisy gradients. Noisy gradients with high variance can cause the SGD algorithm to converge slowly or diverge [2]. The subchains should also be sampled randomly without replacement until all subchains in $\mathbf{y}$ are depleted to speed up convergence. To ensure unbiasedness of the noisy gradients, the gradients computed in each iteration need to be multiplied by a batch factor

$$c = \frac{T - \Delta t}{n_{minibatch}} \tag{10}$$

where $n_{minibatch}$ is the number of subchains in each mini-batch. The scaled noisy gradients can then be used by SGD algorithm of choice to optimise $\mathbb{L}$. In our implementation of the algorithm, gradients are computed using the Python automatic differentiation tool [17] and the optimisation is performed using Rmsprop [22].

## 4 Related Work

Copulas have previously been adapted in variational inference literatures as a tool to model posterior dependency in models with i.i.d. data assumption [23, 10]. However, the previously proposed approaches cannot be directly applied to HMM class of models without addressing parameter estimation issues as the dimensionality of the posterior distributions grow with the length of sequences. The proposed formulation of the variational distribution circumvents the problem by exploiting the chain structure of the model, coupling only random variables within the same chain that are adjacent in time with a bivariate Gaussian-Bernoulli copula, leading to a coherent chain of bivariate Gaussian copulas as the variational distribution.

On the other hand, a stochastic variational inference algorithm that also aims to scale HMM class of models to long sequences has previously been proposed in [5]. Our proposed algorithm differs from the existing approach in that it does not require explicit message passing to perform inference and learning. Applying the algorithm proposed in [5] to FHMM requires multiple message passing iterations to determine the buffer length of each subchain in the mini batch of data, and the procedure needs to be repeated for each FHMM Markov chain. The message passing routines can be expensive as the number of Markov chains grows. In contrast, the proposed recognition network approach eliminates the need for iterative message passing and allows the variational distributions to be learned directly from the data using gradient descent. The use of recognition networks also allows fast inference at run-time with modern parallel computing hardwares.

The use of recognition networks as inference devices for graphical models has received much research interest recently because of its scalability and simplicity. Similar to our approach, the algorithms proposed in [4, 13] also make use of the recognition networks for inference, but still rely on message passing to perform certain computations. In addition, [1] proposed an inference algorithm for state space models using a recognition network. However, the algorithm cannot be applied to models with non-Gaussian posteriors.

Finally, the proposed algorithm is analogous to composite likelihood algorithms for learning in HMMs in that the data dependency is broken up according to subchains to allow tractable computations [6]. The EM-composite likelihood algorithm in [6] partitions the likelihood function according to subchains, bounding each subchain separately with a different posterior distribution that uses only the data in that subsequence. Our recognition models generalize that.

## 5 Experiments

We evaluate the validity of our algorithm and the scalability claim with experiments using real and simulated data. To validate the algorithm, we learn FHMMs on simulated and real data using

the proposed algorithm and the existing SMF-EM algorithm. The models learned using the two approaches are compared with log-likelihood (LL). In addition, we compare the learned FHMM parameters to parameters used to simulate the data. The validation experiments ensure that the proposed approach does not introduce further approximation bias compared to SMF.

To verify the scalability claim, we compare the LL of FHMMs with different numbers of hidden chains learned on simulated sequences of increasing length using the proposed and SMF-based EM algorithms. Two sets of experiments are conducted to showcase scalability with respect to sequence length and the number of hidden Markov chains. To simulate real-world scenarios where computing budget is constrained, both algorithms are given the same fixed computing budget. The learned FHMMs are compared after the computing budget is depleted. Finally, we demonstrate the scalability of the proposed algorithm by learning a 10 binary hidden Markov chains FHMM on long time series recorded in a real-world scenario.

## 5.1 Algorithm Validation

**Simulated Data** We simulate a $1,000$ timesteps long 2-dimensional sequence from a FHMM with 2 hidden binary chains and Gaussian emission, and attempt to recover the true model parameters with the proposed approach. The simulation procedure is detailed in the *Supplementary Material*. The proposed algorithm successfully recovers the true model parameters from the simulated data. The LL of the learned model also compared favorably to FHMM learned using the SMF-EM algorithm, showing no visible further bias compares to the proven SMF-EM algorithm. The LL of the proposed algorithm and SMF-EM are shown in Table 1. The learned emission parameters, together with the training data, are visualised in Figure 1.

**Bach Chorales Data Set** [16] Following the experiment in [8], we compare the proposed algorithm to SMF-EM based on LL. The training and testing data consist of 30 and 36 sequences from the Bach Chorales data set respectively. FHMMs with various numbers of binary hidden Markov chains are learned from the training data with both algorithms. The log-likelihoods, tabulated in Table 1, show that the proposed algorithm is competitve with SMF-EM on a real data set in which FHMM is proven to be a good model, and show no further bias. Note that the training log-likelihood of the FHMM with 8 chains trained using the proposed algorithm is smaller than the FHMM with 7 chains, showing that the proposed algorithm can be trapped in bad local minima.

## 5.2 Scalability Verification

**Simulated Data** This experiment consists of two parts to verify scalability with respect to sequence length and the state space size. In the first component, we simulate 2-dimensional sequences of varying length from a FHMM with $4$-binary chains using an approach similar to the validation experiment. Given fixed computing budget of 2 hours per sequence on a 24 cores Intel i7 workstation, both SMF-EM and the proposed algorithm attempt to fit 4-chain FHMMs to the sequences. Two testing sequences of length $50,000$ are also simulated from the same model. In the second component, we keep the sequence length to $15,000$ and attempt to learn FHMMs with various numbers of chains with computing budget of $1,000$s. The computing budget in the second component is scaled according to the sequence length. Log-likelihoods are computed with the last available learned parameters after computing time runs out. The proposed algorithm is competitive with SMF-EM when sequences are shorter and state space is smaller, and outperforms SMF-EM in longer sequences and larger state space. The results in Figure 2 and Figure 3 both show the increasing gaps in the log-likelihoods as sequence length and state space size increased. The recognition networks in the experiments have 1 hidden layer with 30 $tanh$ hidden units, and rolling window size of 5. The marginal and correlation recognition networks for latent variables in the same FHMM Markov chain share hidden units to reduce memory and computing requirements as the number of Markov chains increases.

**Household Power Consumption Data Set** [16] We demonstrate the applicability of our algorithm to long sequences in which learning with SMF-EM using the full data set is simply intractable. The power consumption data set consists of a 9-dimensional sequence of $2,075,259$ time steps. After dropping the date/time series and the current intensity series that is highly correlated with the power consumption series, we keep the first $10^6$ data points of the remaining 6 dimensional sequence for training and set aside the remaining series as test data. A FHMM with 10 hidden Markov chains is

learned on the training data using the proposed algorithm. In this particular problem, we force all 20 recognition networks in our algorithm to share a common tanh hidden layer of 200 units. The rolling window size is set to 21 and we allow the algorithm to complete $150,000$ SGD iterations with 10 subchains per iteration before terminating. To compare, we also learned the 10-chain FHMM with SMF-EM on the last $5,000$ data points of the training data. The models learned with the proposed algorithm and SMF-EM are compared based on the Mean Squared Error (MSE) of the smoothed test data (i.e., learned emission means weighted by latent variable posterior). As shown in Table 2, the test MSEs of the proposed algorithm are lower than the SMF-EM algorithm in all data dimensions. The result shows that learning with more data is indeed advantageous, and the proposed algorithm allows FHMMs to take advantage of the large data set.

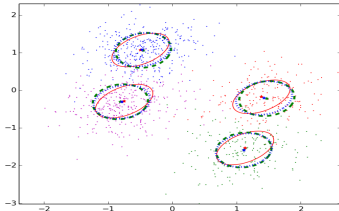 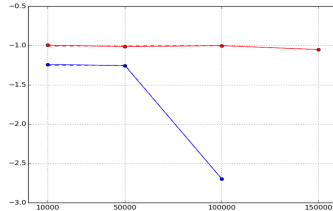 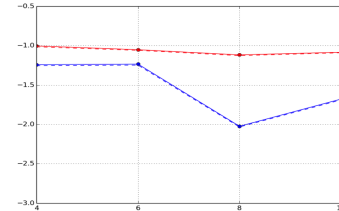

Figure 1: Simulated data in validation experiments with the emission parameters from simulation (red), learned by proposed algorithm (green) and SMF-EM (blue). The emission means are depicted as stars and standard deviations as elliptical contour at 1 standard deviation.

Figure 2: The red and blue lines show the train (solid) and test (dashed) LL results from the proposed and SMF-EM algorithms in the scalability experiments as the sequence length ($x$-axis) increases. Both algorithms are given 2hr computing budget per data set. SMF-EM failed to complete a single iteration for length of $150,000$.

Figure 3: The red and blue lines show the train (solid) and test (dashed) LL results from the proposed and SMF-EM algorithms in the scalability experiments as the number of hidden Markov chain ($x$-axis) increases. Both algorithms are given $1,000$s computing budget per data set.

| $n_{chain}$ | Proposed Algo. | | SMF | |
|---|---|---|---|---|
| | $LL_{train}$ | $LL_{test}$ | $LL_{train}$ | $LL_{test}$ |
| | Simulated Data | | | |
| 2 | -2.320 | -2.332 | -2.315 | -2.338 |
| | Bach Chorales | | | |
| 2 | -7.241 | -7.908 | -7.172 | -7.869 |
| 3 | -6.627 | -7.306 | -6.754 | -7.489 |
| 4 | -6.365 | -7.322 | -6.409 | -7.282 |
| 5 | -6.135 | -6.947 | -5.989 | -7.174 |
| 6 | -5.973 | -6.716 | -5.852 | -7.008 |
| 7 | -5.754 | -6.527 | -5.771 | -6.664 |
| 8 | -5.836 | -6.722 | -5.675 | -6.697 |

Table 1: LL from the validation experiments. The results demonstrate that the proposed algorithm is competitive with SMF. Plot of the Bach chorales LL is available in the *Supplementary Material*.

| Dim. | $MSE_{SMF}$ | $MSE_{Proposed}$ |
|---|---|---|
| 1 | 0.155 | 0.082 |
| 2 | 0.084 | 0.055 |
| 3 | 0.079 | 0.027 |
| 4 | 0.466 | 0.145 |
| 5 | 0.121 | 0.062 |
| 6 | 0.202 | 0.145 |

Table 2: Test MSEs of the SMF-EM and the proposed algorithm for each dimension in the household power consumption data set. The results show that the proposed algorithm is able to take advantage of the full data set to learn a better model because of its scalability. Plots of the fitted and observed data are available in the *Supplementary Material*.

## 6 Conclusions

We propose a novel stochastic variational inference and learning algorithm that does not rely on message passing to scale FHMM to long sequences and large state space. The proposed algorithm achieves competitive results when compared to structured mean-field on short sequences, and outperforms structured mean-field on longer sequences with a fixed computing budget that resembles a real-world model deployment scenario. The applicability of the algorithm to long sequences where the structured mean-field algorithm is infeasible is also demonstrated. In conclusion, we believe that the proposed scalable algorithm will open up new opportunities to apply FHMMs to long sequential data with rich structures that could not be previously modelled using existing algorithms.

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
