[Supplementary Material · ahmm-supplement-cameraready-v3.pdf]

# Supplementary Material
# Scaling Factorial Hidden Markov Models: Stochastic Variational Inference without Messages

**Yin Cheng Ng**
Dept. of Statistical Science
University College London
y.ng.12@ucl.ac.uk

**Pawel Chilinski**
Dept. of Computing Science
University College London
ucabchi@ucl.ac.uk

**Ricardo Silva**
Dept. of Statistical Science
University College London
r.silva@ucl.ac.uk

This document contains additional experimental results, methodologies and derivations that supplement the main text of the paper.

## 1 Supplementary Experimental Results

The following two subsections highlight the importance of capturing the posterior dependency within each Markov chain, and present plots that supplement the results in the *Experiment* section of the main text respectively.

### 1.1 Importance of Posterior Dependency

We highlight the importance of capturing posterior dependency by comparing FHMMs learned using the proposed algorithm and a modification in which the copula correlation $\rho_t$ is always set to 0. The FHMMs learned using the proposed algorithm and its modified version have training LL of -2.284 and -2.480 respectively, illustrating the importance of the posterior dependency. The transition matrices learned by the proposed algorithm and its modification yielded errors of 0.036 and 0.548 respectively when compared to the true transition matrices that generated the data, using Frobenius norm of the residual matrix as error measure.

### 1.2 Supplementary Plots

Figure 1: The figure shows the log-likelihood for the Bach chorales data set as reported in *Table 1* of the main text.

Figure 2: The figure shows a 5000 timestep-long segment of the 6 dimensional Household Power Consumption test data set (gray) together with the corresponding smoothed data from the SMF-EM (blue) algorithm and the proposed algorithm (red). The MSEs are reported in *Table 2* of the main text.

## 2 Synthetic Data Simulation Methodology

We simulate synthetic data to verify the accuracy of the proposed algorithm, and to compare the scalability of the proposed algorithm and the SMF-EM algorithm. The synthetic data is simulated from FHMMs with various numbers of Markov chain and 2 dimensional Gaussian emission distribution. The parameters of the FHMMs are randomly sampled from various probability distributions as specified in the following algorithm, with additional transformations to control the scale of random noise in the simulated data. Every simulated data set consists of 3 training sequences and 2 test sequences.

The data simulation algorithm from a FHMM with $M$ binary Markov chains and a $D$ dimensional Gaussian emission distribution is as follow:

1. Sample emission mean weight matrix $\mathbf{W} \in \mathbb{R}^{(M+1) \times D}$ from $Gaussian(\mathbf{0}, \mathbb{I})$

2. For Markov chain $m = 1 \dots M$:

    (a) Sample a matrix $\mathbf{P} \in \mathbb{R}^{2 \times 2}$ from $Uniform(0, 1)$

    (b) Normalize $\mathbf{P}$ such that $\sum_{i,j}^{2} P_{i,j} = 1$

    (c) Initial distribution $p(s_1^m = 1) = P_{1,0} + P_{1,1}$

    (d) Transition probability $p(s_t^m = 1 | s_{t-1}^m = 0) = \frac{P_{1,0}}{1 - P_{0,1} - P_{1,1}}$

    (e) Transition probability $p(s_t^m = 1 | s_{t-1}^m = 1) = \frac{P_{1,1}}{P_{0,1} + P_{1,1}}$

3. Sample emission covariance matrix $\mathbf{C} \in \mathbb{R}^{D \times D}$ from $Wishart(\mathbb{I}, D + 1)$

4. Rescale the $\mathbf{W}$ matrix to control the amount of noise:

    (a) Compute variance of the Bernoulli random variable in each Markov chain $\mathbf{v} = [v_1, \dots, v_M]^T$

    (b) Compute signal standard deviation per Markov chain $\sigma_{\mathbf{s}} = \sqrt{diag(\widetilde{\mathbf{W}}^T \mathbf{V} \widetilde{\mathbf{W}})}$ where $\widetilde{\mathbf{W}} \in \mathbb{R}^{M \times D}$ is the first $M$ columns of $\mathbf{W}$ and $\mathbf{V}$ is a square matrix with $\mathbf{v}$ on the diagonal and $0$ everywhere else

    (c) Rescale the first $M$ columns of $\mathbf{W}$ such that $\mathbf{W}_{:,\mathbf{i}} = \frac{\mathbf{W}_{:,\mathbf{i}}}{s\sigma_{s(i)}}$ where $s$ is a pre-specified signal to noise ratio and $\sigma_{s(i)}$ is the $i^{th}$ element of $\sigma_{\mathbf{s}}$

# 3 Variational Chains of Bivariate Gaussian Copulas Parametrization

In this section we show the parametrization of the approximate posterior:

$$q(\mathbf{s}) = \prod_{m=1}^{M} q(\mathbf{s}^m) \tag{1}$$

$$q(\mathbf{s}^m) = \frac{\prod_{t=2}^{T} q(s_{t-1}^m, s_t^m)}{\prod_{t=2}^{T-1} q(s_t^m)} \tag{2}$$

We assume marginals of the hidden variables to be the Bernoulli distribution:

$$q(S_t^m = s_t^m) = \begin{cases} 1 - \theta_{t,m}, & s_t^m = 0 \\ \theta_{t,m}, & s_t^m = 1 \end{cases} \tag{3}$$

The parameters $\theta_{t,m}$ are encoded as the function of the data as:

$$\begin{aligned} \theta_{t,m} &= f_\theta^m(\Delta \mathbf{y}_t) \\ \Delta \mathbf{y}_t &= (y_{t-\frac{1}{2}\Delta t}, \ldots, y_t, \ldots, y_{t+\frac{1}{2}\Delta t}) \end{aligned} \tag{4}$$

We parametrize the joint between adjacent hidden variables in the same chain with the Gaussian copula:

$$q(S_t^m = 0, S_{t-1}^m = 0) = \Phi_2(\Phi^{-1}(1 - \theta_{t,m}), \Phi^{-1}(1 - \theta_{t-1,m}), \rho_{t,m}) = q_{00_t}^m \tag{5}$$

where:

- $\Phi_2$ is standardized bivariate normal CDF.
- $\Phi^{-1}$ is inverse of standardized univariate normal CDF.
- $\rho_{t,m}$ is the correlation of the copula which is parametrized according to

$$\rho_{t,m} = f_\rho^m(\Delta \mathbf{y}_t) \tag{6}$$

We can write the entire discrete joint distribution $q(S_t^m, S_{t-1}^m)$ with respect to $q(S_t^m = 0, S_{t-1}^m = 0)$:

$$\begin{aligned} q(S_t^m = 0, S_{t-1}^m = 0) &= q(S_t^m \le 0, S_{t-1}^m \le 0) = q_{00_t}^m \\ q(S_t^m = 1, S_{t-1}^m = 0) &= q(S_t^m \le 1, S_{t-1}^m \le 0) - q(S_t^m \le 0, S_{t-1}^m \le 0) \\ &= q(S_{t-1}^m \le 0) - q_{00_t}^m = 1 - \theta_{t-1,m} - q_{00_t}^m \\ q(S_t^m = 0, S_{t-1}^m = 1) &= q(S_t^m \le 0, S_{t-1}^m \le 1) - q(S_t^m \le 0, S_{t-1}^m \le 0) \\ &= q(S_t^m \le 0) - q_{00_t}^m = 1 - \theta_{t,m} - q_{00_t}^m \\ q(S_t^m = 1, S_{t-1}^m = 1) &= q(S_t^m \le 1, S_{t-1}^m \le 1) - q(S_t^m \le 1, S_{t-1}^m \le 0) \\ &\quad - q(S_t^m \le 0, S_{t-1}^m \le 1) + q(S_t^m \le 0, S_{t-1}^m \le 0) \\ &= 1 - q(S_{t-1}^m \le 0) - q(S_t^m \le 0) + q_{00_t}^m \\ &= \theta_{t,m} + \theta_{t-1,m} - 1 + q_{00_t}^m \end{aligned} \tag{7}$$

# 4 Computing Variational Distributions at the Edges

This section describes the algorithm to compute the variational posterior distributions of Markov chain random variables that are close to the beginning and the end of the timesteps. The variational distributions of these random variables cannot be computed by the recognition neural networks, as their corresponding input sequences are out of range. The algorithm is adapted from the standard structured mean-field (SMF) algorithm for FHMMs.

Given sequential data $\mathbf{Y} = \{\mathbf{y_1}, \ldots, \mathbf{y_T}\}$ of length $T$ and recognition neural networks with rolling window size $\Delta T + 1$, the recognition neural networks can only compute posterior distributions of random variable $s_t^m$ in Markov chain $m$ up to timestep $t = T - \frac{\Delta T}{2}$. The variational posterior distributions for the remaining sequence of random variables $q(s_{T-\frac{\Delta T}{2}+1}^m, \ldots, s_T^m)$ needs to be parameterized and computed as follow if required.

Following the standard SMF Markov chain variational distribution parameterization, the variational distributions for $s_{T-\frac{\Delta T}{2}}^m, \ldots, s_T^m$ can be written as

$$q(s_{T-\frac{\Delta T}{2}}^m, \ldots, s_T^m) = q(s_{T-\frac{\Delta T}{2}}^m) \prod_{t=T-\frac{\Delta T}{2}}^{T-1} q(s_{t+1}^m | s_t^m) \tag{8}$$

$$q(s_{t+1}^m | s_t^m) = (h_{t+1,0}^m {P_{0,0}^m}^{1-s_t^m} {P_{0,1}^m}^{s_t^m})^{1-s_{t+1}^m} (h_{t+1,1}^m {P_{1,0}^m}^{1-s_t^m} {P_{1,1}^m}^{s_t^m})^{s_{t+1}^m} \tag{9}$$

where $P_{i,j}^m$ is the $i,j^{th}$ element of the $m^{th}$ Markov chain transition matrix, $q(s_{T-\frac{\Delta T}{2}}^m)$ is computed by the recognition neural networks, and $\mathbf{h}_{t+1}^m = [h_{t+1,0}^m, h_{t+1,1}^m]^T$ are parameters with the following fixed point equations.

$$\mathbf{h}_{t+1}^m = exp(\mathbf{W}_{:,m}^T \mathbf{C}^{-1} \widetilde{\mathbf{y}}_{t+1}^m - \frac{1}{2}\gamma^m) \tag{10}$$

$$\gamma^m = diag(\mathbf{W}_{:,m}^T \mathbf{C}^{-1} \mathbf{W}_{:,m}) \tag{11}$$

$$\widetilde{\mathbf{y}}_{t+1}^m = \mathbf{y}_t - \sum_{l \neq m} \mathbf{W}_{:,l} < s_{t+1}^l > \tag{12}$$

The following forward-backward recursions can be used to compute the expectations $< s_{t+1}^l >$,

$$\alpha_{t+1}^m(s_{t+1}^m) = \sum_{s_t^m} \alpha_t^m(s_t^m) P(s_{t+1}^m | s_t^m) h_{t+1,s_{t+1}^m}^m \tag{13}$$

$$\alpha_{T-\frac{\Delta T}{2}}^m(s_{T-\frac{\Delta T}{2}}^m) = q(s_{T-\frac{\Delta T}{2}}^m)$$

$$\beta_t^m(s_t^m) = \sum_{s_{t+1}^m} \beta_{t+1}^m(s_{t+1}^m) P(s_{t+1}^m | s_t^m) h_{t+1,s_{t+1}^m}^m \tag{14}$$

$$\beta_T^m(s_T^m) = 1$$

$$< s_t^m > \propto \alpha_t^m(s_t^m) \beta_t^m(s_t^m) \tag{15}$$

The variational distributions for random variables at the beginning of the sequence can be computed similarly, with $\alpha_1^m(s_1^m) = 1$ and $\beta_{\frac{\Delta T}{2}+1}^m(s_{\frac{\Delta T}{2}+1}^m)$ computed by the recognition neural network.