[Reviews · NeurIPS 2016]

Reviewer 1

Summary

This paper presents an approach to variational inference and learning in factored hidden Markov models (FHMMs) specifically tailored to long sequences. The approach combines several lines of related work including the use of a variational posterior distribution based on a chain of bivariate Gaussian copula models for each dimension of the latent space, the use of feed-forward recognition networks to predict variational parameters, and the use of temporal mini-batches and stochastic variational inference for learning. The authors show that the proposed approach achieves log likelihood results that are comparable to the existing structured mean field (SMF) approach on short sequences and small latent space sizes, while the proposed approach is able to scale to much larger sequences.

Qualitative Assessment

Technical quality: This paper tackles the problem of inference and learning of factored HMMs on large sequences and with large latent dimensionality. The primary contribution of the paper is integrating several existing approaches together to enable large-scale learning of FHMMs without a loss in modeling performance. The technical details of the components of the approach (the bivariate Gaussian copula variation posterior, the recognition network, the SVI learning approach) appear to be technically correct. The experimentation touches on the correct points including the accuracy of the learned models and the scalability of the proposed approach. The accuracy of the learned models is assessed using log likelihood on held-out test data. The experiments show that the model performs similarly to the SMF approach on both simulated and real (the Bach Corals) data. The speed-accuracy trade-off of the proposed approach relative to SMF is assessed by looking at the log likelihood obtained under a fixed computation time budget. Here the proposed method outperforms SMF. A weak point in the experiments are that no standard errors are included and no hypothesis tests are conducted to establish statistical significance of the log likelihood differences. It would also have been nice to see timing comparisons on the Bach data set instead of just the synthetic data. The application to the power consumption data set is a nice demonstration of the scalability of the proposed approach, but as the results are purely qualitative, they do not add much else to the paper. Novelty/originality: As mentioned previously, this paper represents a novel combination of a number of prior approaches to provide faster inference and learning for the FHMM model class in the sense that these components have not been combined previously in the manner presented. However, the combination is relatively straightforward and no major adaptations of the components was required. On the other hand, the proposed approach is able to deal with data sets of much larger sizes than previous approaches, enabling novel data analysis capabilities. Impact: The work does have the potential for strong impact in areas using factored HMMs for data analysis as the proposed methodology allows for fitting more complex models to larger data sets. The impact relative to newer deep learning models like Deep Temporal Sigmoid Belief Networks is less clear and is not discussed by the authors. Clarity and presentation: The grammar in the paper is off in many places, but this does not affect ability to understand the technical content of the paper. The figures could use some work such as replacing figures 1,2,3 with proper sub-figures, as well as adjusting the axis tick and label fonts so they are readable in print.

Confidence in this Review

2-Confident (read it all; understood it all reasonably well)


Reviewer 2

Summary

The authors present a new algorithm for inference in factorial hidden Markov models. The novelty in their algorithm is a combination of a) stochastic variational inference, b) amortised inference using neural nets, c) gaussian copulas for modelling dependencies in the temporal dimenions. I am not hugely familiar with Gaussian copulas but one of my main questions for the authors is why the need the amortised inference neural net predicts the parameters of a Gaussian copula rather than some "simpler" parameterisation of the 2x2 transition matrix? Minor comments: - 31: [4][8] -> [4,8] - 39: [12][10] -> [12,10] - 39: i.i.d -> i.i.d. - eq1: maybe use some brackets to make clear which product signs apply to which terms. - 89: [16][2] -> [16,2] (and following ...)

Qualitative Assessment

The paper presents a novel combination of existing techniques with a good scope for applications. The authors performed a satisfying experimental evaluation. Particularly the experiment on the household power consumption dataset shows that this algorithm is particularly useful in practice.

Confidence in this Review

2-Confident (read it all; understood it all reasonably well)


Reviewer 3

Summary

The paper proposes a variational inference strategy for discrete-state factorial hidden Markov models (FHMMs) that uses recognition networks to replace the standard FHMM message-passing (and possibly iterative) inference steps. Inference proceeds by sampling subsequence windows of fixed length, applying a recognition network to yield Gaussian copula parameters for a variational factor on states, and then using the expected sufficient statistics under this variational distribution for updating the parameters of the generative model and recognition network. The method is evaluated on some simulated and real datasets and compared to the batch structured mean field (SMF) algorithm.

Qualitative Assessment

The basic idea of this paper may be a good one, but the paper itself does not adequately defend the idea or compare it to the right baselines. The main selling point is that the proposed inference strategy does not require HMM message passing as would be needed for the standard structured mean field (SMF) approach to factorial HMMs (FHMMs). The computational burden of message passing with long sequences is called "insurmountable" and "enormous" (Section 2.1.1). However, as noted in Section 2.1.1, the time and space complexity of the HMM message passing algorithm is linear in the (sub)sequence length T, and using the techniques of [4] this subsequence could be the same length as the window size used in the proposed method. Because the proposed method also requires time that is at least linear in the window length (it applies a recognition network to the full window of data), the basic proposition of improved scalability is not so clear. Furthermore, using message passing has the advantages of adapting to unobserved data and varying sequence lengths (as outlined nicely on lines 193-197). Adapting the techniques of [4] to the FHMM setting would require iterative inference for every minibatch of data; that is, message passing would have to be run iteratively for each component chain until the factors converge. Avoiding this iterative inference with recognition networks would be a compelling story. However, that is not the line of argument presented in the current paper, which instead only claims that message passing itself is prohibitive. More importantly, since there is no clear a priori advantage of the proposed method over a scheme using [4], any advantage would need to be demonstrated empirically. However, the experiments in the current paper do not compare to [4], and instead only compare to SMF-EM, which sounds like the batch version of the method. The paper does point out that the "lack of message passing procedure allows subchains from the long sequence to be distributed across a network of computers" (lines 249-250), which could yield an advantage (though the stochastic gradients used in [4] could also be computed in parallel for a given minibatch), but it would also need to be demonstrated. The remainder of my comments highlights some minor ideas for improvement. Lines 40-41 suggest that the use of recognition networks in non-factorized variational posteriors has been limited, but there are several works that could be cited here. In particular, you could cite "Composing graphical models with neural networks for structured representations and fast inference" by Johnson et al., "Black box variational inference for state space models" by Archer et al., and "Deep Kalman filters" by Krishnan et al. Lines 23-24 imply that KL(q||p) can be computed, but that is not true; indeed, because the marginal likelihood log p(y) can be written as a sum of KL(q||p) and the ELBO (i.e. the variational lower bound), and because we can evaluate the ELBO, it is not possible to compute KL(q||p) without also being able to compute the marginal likelihood log p(y). The sentence could be rephrased to say "allowing tractable computations of the gradient of the KL divergence". On line 31, might also want to cite Johnson and Willsky (2014), which considered SVI for HMMs slightly earlier, but if you're just only going to cite one then Foti et al. (2014) is probably better for this paper (because of line 105). English error on line 110: "Recognition neural network was initially proposed ..." Why not use a bidirectional RNN architecture, like in CTC? Then information can flow like message passing but you can still avoid having to do iterative updates and message passing for each chain separately. Maybe Table 1 should be a plot.

Confidence in this Review

2-Confident (read it all; understood it all reasonably well)


Reviewer 4

Summary

This paper combines ideas from the Gaussian copula literature and recent advances in stochastic variational inference and recognition neural networks to avoid message passing in factorial hidden Markov models. The idea works as follows: -learn variational distributions of bivariate Gaussian copulas for the state transitions within a single hidden Markov chain. -instead of learning all the variational parameters needed, learn parameters for a recognition model. Compute the variational parameters for the Gaussian copula as the output of the recognition model, where the inputs are a rolling window of observed data. They then setup the ELBO and show how to optimize it.

Qualitative Assessment

This is a clever paper. It combines several recent ideas and applies them in a simple but non-obvious way to an old problem. It is well written, and despite using methods from several different areas, it explains what is needed and is mostly self-contained. The experimental results are convincing. My one technical issue: what is the difference between the ELBO approximation used in (9) and the true ELBO? It would be good if that was explained, particularly if the terms that are different can be isolated and some intuitive justification can be provided for why they should be small. Is it only that we lose the very early terms and the end terms of the chain, and thus for a sufficiently long term it shouldn't matter? I have a few questions: -the emissions distributions all have the same covariance matrix. Is this standard for FHMMs? -are you the first to use Gaussian copulas for transition probabilities when states are indicator variables? There are a couple typos/grammatical issues that I noticed: -on line 160, ramdon should be random -line 180, adapt should be adapted I have the following suggestions: -theoretical analysis of the complexity improvements from avoiding message passing. This isn't entirely clear. -a more convincing discussion of why FHMMs are important today

Confidence in this Review

2-Confident (read it all; understood it all reasonably well)


Reviewer 5

Summary

This paper develops more scalable variational methods for factorial hidden Markov models by combining ideas from structured mean-field optimization, recognition networks and stochastic gradient descent. In the proposed approximating posterior based on a Gaussian bivariate copula, each timestep t within chain m has two variational parameters: marginal probability \theta_tm and correlation \rho_tm. These two parameters, together with parameters of neighboring timesteps, can represent any possible pairwise distribution for neighboring binary states. For a single binary chain of length T, there are in total 2T required parameters. The choice of the marginal and correlation values, as opposed to other representations, is done to make each value predictable from a separate recognition network. The proposed recognition network approach reduces the total number of parameters per chain from 2T to 2H, where H is the total number of parameters in a feed-forward network that predicts the scalar correlation or marginal probabilty value for timestep t given a fixed-width window of data around timestep t. The length of the fixed-width window and the internal structure of the recognition network are in general application dependent and likely must be chosen carefully. The whole approach results in a variational objective function parameterized by global parameters like emission and transition probabilities, and the recognition network weight vectors which control the local state sequence posterior. The number of parameters per sequence is much less than the structured mean field approach, leading to gains in training and evaluation speed. This function can be optimized via stochastic gradient descent, where we repeatedly sample small windows of small, fixed length from the long sequence and take gradient steps. Automatic differentiation packages are used here. Experiments compare the proposed approach against a full-batch SMF baseline on several datasets including Bach chorales and household power consumption.

Qualitative Assessment

Overall I think this work is interesting and the proposed union of recognition networks and variational inference is of growing interest in the community. This could make a good poster presentation at NIPS, but I could also understand arguments for borderline rejection. My primary concerns are the method requires lots of tuning to find the right recognition network structure with little guidance on this front, and that the experiments are a bit weak in that they only compare the proposed method to a very basic method (full-batch structured mean field) without evaluating recent alternatives like Foti et al [4] which might be more scalable by using small windows of the long sequence without the added complexity of recognition networks. Technical feedback ================== Overall the proposed method requires several important decisions on the part of the end user: recognition network structure, fixed-window length, how to handle the edge case timesteps at the beginning and end of the sequence where the network's window does not fit, etc. These are casually mentioned, but no experiment really provides guidance about how to deal with these in a principled way, or even compares tradeoffs in runtime vs accuracy under different possibilities. I think this lack of careful consideration will make this method difficult to apply in practice, and I'd like to see more discussion in a revision. The question of local optima is definitely relevant here, and little guidance is provided on this front. Without showing results from many independent initializations, it is difficult to know whether the proposed approach might be more or less vulnerable to local optima than the standard SMF approach. Is code available? Perhaps sharing code would mitigate some concerns, especially when for dealing with the edge cases. I'd like to see some scalability experiments that directly compare two methods for a sequence of length T and a trained model with a single binary chain: (1) using a recognition network of width W to predict the entire state sequence, including edge cases, and (2) using the standard forward/backward algorithm to estimate the state sequence. I think it would be useful to directly quantify how much easier this kind of test-time inference gets under the proposed approach... is it 2x? 10x better? Hard to say right now. Experiment feedback =================== Looking at Table 1 for the Bach dataset, the baseline structured mean field (SMF) approach produces log likelihood on training (LLtrain) values that continue to improve with more hidden chains all the way through 8 chains, as we would expect because the model capacity is improving and can fit the data better. However, for the proposed algorithm, the LLtrain with 7 chains (-5.75) is noticeably better than for 8 chains (-5.836), which I would not expect. I would guess this is a local optima issue. I'd like to hear more about this in rebuttal. Should we be concerned that for large model capacity the performance of the proposed approach might suffer? This doesn't really reassure me that the performance is always indistinguishable from SMF. Overall, I'd much rather see Table 1 turned into a line plot of performance vs # chains. Tables are not ideal for helping your audience understand relative performance trends. When evaluating scalability in Sec. 5, the comparison between the proposed approach (which can stochastically use subsequences *and* take advantage of the cheap recognition network) and the SMF-EM algorithm, which must process a whole sequence before each global parameter update, might not be the fairest possible comparison. I'd be curious how the Foti et al method [4], which can process small subsequences from the long sequence but does *not* use the recognition network, stacks up in this comparison. This may be too much to ask, but otherwise this experiment isn't especially convincing that each piece in the proposed pipeline lead to scalability wins. It's hard to know whether the effort for the recognition network is really "worth it", or whether just doing stochastic updates after a few short windows would give similar performance. Household power dataset questions: how long (wallclock time) did training take here? How many SGD iterations were done? How much data does one iteration use? Instead of just showing the predictions of the method on the power dataset, I'd suggest a within-method comparison, where you choose some useful tweak to your model (5 vs 10 chains, or rolling window size of 5 vs 10, etc) and report how the resulting performance vs speed tradeoffs. Typos, etc ========== line 160 typo: "ramdom variables in different chains" Sec 5.2 suggestion: Say "this experiment consists of two parts" instead of "two components".

Confidence in this Review

2-Confident (read it all; understood it all reasonably well)


Reviewer 6

Summary

The authors propose an efficient approach to perform inference and learning in Factorial Hidden Markov Models (FHMMs) for long time-series that uses copulas to model the approximating posteriors and neural networks to perform inference.

Qualitative Assessment

I think that the paper should be reformulated to focus more on crucial aspects related to the contributions: I find that there is too much repeated general explanation on the class of models and inference considered, and not enough information about the specific modeling, approximation and inference choices. I would not put much emphasis on the fact that a completely factorized mean-field approach is poor for time-series. Fully factorized approaches over time-steps were used only when variational methods were first utilized for time-series, but no one would consider such approaches reasonable any more. I would therefore put Section 5.3 in the additional material. I would explain much more in detail the reasons for considering a copula model. What are the properties of copulas? How well do the authors think they can model the approximating posterior? In the experimental section, the simulation is done with just 2 chains? I guess, with more chains there could be different parametrizations that give the same likelihood, and therefore it would be difficult to show that the correct parameters are found by the model. Is this the reason for considering such a simple scenario? This is unfortunate as we can't really understand how the model would do in more difficult cases. Performance could be evaluated on how well the model retrieves the correct hidden states instead, so that it would make sense to use a high number of chains. I am not fully convinced about the usefulness of specifically addressing long time-series from the experimental section. How would the FHMM perform on the power consumption data set in which the 1 million long time-series is segmented, say, into 5 subsequences? In conclusion, we are experiencing an explosion of papers that are simple variants on the idea of stochastic variational inference/stochastic backpropagation, using neural networks. Due to their simplicity from a mathematical viewpoint, I think that the main value of such types of papers is in the experimental evaluation. I think, however, that the experimental section of this paper is not very strong. I therefore struggle to see how the community could benefit from its publication.

Confidence in this Review

3-Expert (read the paper in detail, know the area, quite certain of my opinion)